Flower-mediated plant-butterfly interactions in an heterogeneous tropical coastal ecosystem

Martínez-Adriano Cristian A. cristian.martinez.cama@gmail.com
Díaz-Castelazo Cecilia
Aguirre-Jaimes Armando
Red de Interacciones Multitróficas, Instituto de Ecología, A. C. , Xalapa , Veracruz , México
Traveset Anna
Electronic publication date: 2018 Sep 7
Publication date: 2018
Volume: 6
Electronic Location ID: e5493
Received 2018 Feb 22; Accepted 2018 Jul 26
Copyright: ©2018 Martínez-Adriano et al.
Copyright year: 2018
Copyright holder: Martínez-Adriano et al.
License: This is an open access article distributed under the terms of the Creative Commons Attribution License, which permits unrestricted use, distribution, reproduction and adaptation in any medium and for any purpose provided that it is properly attributed. For attribution, the original author(s), title, publication source (PeerJ) and either DOI or URL of the article must be cited.
License URL: https://creativecommons.org/licenses/by/4.0/

Keywords: Butterfly diversity, Flower visitor, Flowering plants, Interaction networks, Plant communities

Funding: Consejo Nacional de Ciencia y Tecnología (CONACyT) 230073 Instituto de Ecología, A.C. (INECOL, A.C.) INECOL 2003011143 This study was developed as part of the doctoral studies of Cristian A. Martínez Adriano, with scholarship number 230073 awarded by the Consejo Nacional de Ciencia y Tecnología (CONACyT), at the doctorate program of Instituto de Ecología, A.C. (INECOL, A.C.). Cecilia Díaz-Castelazo was provided financial support for fieldwork through the project 2003011143 by INECOL, A.C. The funders had no role in study design, data collection and analysis, decision to publish, or preparation of the manuscript.

==============================
Background

Interspecific interactions play an important role in determining species richness and persistence in a given locality. However at some sites, the studies, especially for interaction networks on adult butterflies are scarce. The present study aimed the following objectives: (1) determine butterfly species richness and diversity that visit flowering plants, (2) compare species richness and diversity in butterfly-plant interactions among six different vegetation types and (3) analyze the structure of butterfly-flowering plant interaction networks mediated by flowers.

Methods

The study was developed in six vegetation types within the natural reserve of La Mancha, located in Veracruz, Mexico. In each vegetation type, we recorded the frequency of flower visits by butterflies monthly in round plots (of radius 5 m) for 12 months. We calculated Shannon diversity for butterfly species and diversity of interactions per vegetation type. We determined the classic Jaccard similarity index among vegetation types and estimated parameters at network and species-level.

Results

We found 123 species of butterflies belonging to 11 families and 87 genera. The highest number of species belonged to Hesperiidae (46 species), followed by Nymphalidae (28) and Pieridae (14). The highest butterfly diversity and interaction diversity was observed in pioneer dune vegetation (PDV), coastal dune scrub (CDS) and tropical deciduous flooding forest and wetland (TDF-W). The same order of vegetation types was found for interaction diversity. Highest species similarity was found between PDV-CDS and PDV-TDF. The butterfly-plant interaction network showed a nested structure with one module. The species Ascia monuste, Euptoieta hegesia and Leptotes cassius were the most generalist in the network, while Horama oedippus, E. hegesia, and L. cassius were the species with highest dependencies per plant species.

Discussion

Our study is important because it constitutes a pioneer study of butterfly-plant interactions in this protected area, at least for adult butterflies; it shows the diversity of interactions among flowering plants and butterflies. Our research constitutes the first approach (at a community level) to explore the functional role of pollination services that butterflies provide to plant communities. We highlighted that open areas show a higher diversity and these areas shared a higher number of species that shaded sites. In the interaction networks parameters, our results highlighted the higher dependence of butterflies by the flowers on which they feed than vice versa. In conclusion, the plant species (as a feeding resource) seem to limit the presence of butterfly species. Thus, this protected area is highly relevant for Lepidoptera diversity and the interaction between these insects and flowering plants. We suggest that studying plant and butterfly diversity in tropical habitats will provide insight into their interspecific interactions and community structure.

Introduction

Interspecific interactions play an important role in determining species richness and persistence in a given locality (Benadi et al., 2013), as well as providing structure to biotic communities. Mutualistic interactions such as plant–pollinator interactions at a community level are known to provide this structure and stability to biotic communities (Thébault & Fontaine, 2010). There are many studies of plant–pollinator interactions (at a community level) that addressing the composition, structure, and benefits of interacting species; many of them focus on the most frequent or efficient flower visitors, i.e., such as bees (Huang & Giray, 2012; Rosas-Guerrero et al., 2014; Ashworth et al., 2015). However, there are many guilds of flower visitors that affect these species assemblages of pollinators and have effects on the entire ecological community (Rosas-Guerrero et al., 2014; Ashworth et al., 2015).

Butterflies are very common and an important group of flower visitors worldwide and specially in tropical ecosystems (Bawa, 1990). Butterflies as pollination providers in the tropics are considered somewhat effective or solely effective in a few plant species (De Araújo, Quirino & Machado, 2014). This insect group also plays an important role as secondary pollinators when primary pollinators are missing (Rosas-Guerrero et al., 2014; Ashworth et al., 2015) and few studies have addressed this aspect. Often butterflies are seen only as floral visitors providing no net benefit to the plants their visit or they are considered nectar robbers; however, there are some species that could play a functional role as true pollinators (this role often is underestimated).

Butterflies have been one of the groups most appealing as a model for monitoring and biodiversity conservation studies (Narayana, Ramesh & Lakshmi, 2017). This is due to advanced development in the study of its systematics, ecology, and biogeography, which gives to this group a significant relevance (Llorente et al., 1993). However, the study of this group has been developed mainly at the regional level, associated with the generation of regional lists and recognition of rare species (Luis & Llorente, 1993). The lack of ecological knowledge of adult butterflies is also an issue in Mexico, notwithstanding the fact that this insect guild presents a remarkable biological richness in the country, presented in taxonomic varieties, associated vegetation types and a high number of endemisms (Llorente-Bousquets et al., 2014). Mexico presents near 9.3% of Papilionoidea (diurnal butterflies) species of the world (Hernández-Baz et al., 2010; Álvarez García, Ibarra-Vázquez & Escalante, 2016; Luis-Martínez et al., 2016), with 2,049 (Llorente, Luis-Martínez & Vargas, 2006a; Hernández-Baz et al., 2010; Luis-Martínez et al., 2011) to 2,105 (Llorente-Bousquets et al., 2014) species of diurnal butterflies described nationwide. The species richness of Veracruz state alone represents about 6.11% of all butterflies in the world (Llorente et al., 2006b; Hernández-Baz et al., 2010). Veracruz has 57.3% of the species-subspecies, and 79.56% of the genera of Papilionoidea in Mexico (Hernández-Baz et al., 2010), as well as 2,000 species of nocturnal Lepidoptera (moths) of the estimated total of 6,500 species (Hernández-Baz et al., 2010).

The Coastal Research Center of La Mancha (Centro de Investigaciones Costeras La Mancha [thereafter La Mancha]) is located in the central part of Veracruz in Mexico and is managed by the Instituto de Ecología, A.C. For this area, the Lepidoptera are an understudied order, however, they are a biological group of importance as herbivores (Castillo-Guevara & Rico-Gray, 2002; Cuautle & Rico-Gray, 2003; López-Carretero et al., 2014) and as flower visitors (Hernández-Yáñez et al., 2013). It is therefore not only important to determine the butterfly species richness, but also to examine their interactions with the feeding plants (Milne & Milne, 1992), since these interactions could be even higher in number than the richness of the interacting species, given that they are rarely specific (Fishbein & Venable, 1996). It is very common in nature for interspecific interactions to occur as complex interaction networks, where a particular flower visitor may visit several plant species, but with a variable efficiency ranging from nectar robbers to potential pollinators (Stebbins, 1970; Fishbein & Venable, 1996; Waser et al., 1996). Butterfly species could also interact with their feeding plants in such a manner that such interactions could be represented as complex networks.

Ecological network analysis (e.g., of complex networks, such as trophic networks) is an ideal approach by which to understand the dynamics of interspecific interactions within communities and ecosystems. This tool could, therefore, be of value for studying and representing complex biotic systems and their emergent properties (Albert, Jeong & Barabasi, 2000; Strogatz, 2001; Almaas, 2007; Campbell et al., 2011), providing an insight into the assemblage of ecological communities (Arii & Parrott, 2004; Capitán, Cuesta & Bascompte, 2009; Fortuna et al., 2010; Campbell et al., 2011). Communities of plants and their flower visitors can, in general, be represented as a bipartite network (two trophic levels) (Campbell et al., 2011), in which the nodes consist of the butterfly and plant species linked by their feeding interactions. However, studies of plant-butterfly interactions are scarce and even non-existent at the level of complex networks, despite the fact that most species of this group are charismatic and ecologically important as flower visitors for certain plant species. In addition, it is interesting to understand the interactions among butterflies and flowers in a tropical ecosystem because La Mancha is a relict of tropical forest established on a dune system on the Gulf of Mexico coast that constitutes an important migratory corridor for several animal guilds that could include butterflies. It is therefore of interest to study not only butterfly species richness, but also the great diversity of flowers visited in the different vegetation types, since the feeding plant (or plants) of the adult stage of the butterfly species are unknown in most studies (see results of Bivar de Sousa et al., 2016).

Given this background and the potential importance of butterflies for plants (as antagonist (herbivores and robbers of nectar and pollen) and providers of pollination services) in the study area, our general objective was to study the species richness and diversity and the structure of butterfly-plant interactions mediated by flowering plants in a tropical coastal ecosystem. Specifically, we aimed to: (1) characterize butterfly species diversity at the study site, (2) characterize species richness and compare species similarity among the six most representative vegetation types of the study area, (3) determine the diversity of butterfly-plant interactions in these vegetation types, and (4) analyze the community-level structure and parameters of the butterfly-plant interaction network, as well as the structure and parameters for each network per vegetation type.

Considering all vegetation types studied, we expected to find a high diversity of butterfly species in La Mancha reserve. We also expected that sites with less tree cover and higher representativeness of herbaceous species would be more important in terms of the use of floral resources for this group of insects and that the butterflies would display a preference for foraging in open sites. Given the resemblance to a network of mutualistic interactions, we expected the network to be mostly nested and to a lesser extent modular since this system bear many similarities to a mutualistic network of pollination. Additionally, we expected higher dependence of butterflies on plants than plants on butterflies in all interaction networks; this prediction was based on field observations where most of the plant species have more than one group of floral visitors to obtain a successful pollination.

Materials & Methods

Study site

The study was developed within the reserve of La Mancha, located in Veracruz, Mexico (19°36′N, 96°22′W); the reserve (including the field station) is approximately 82.29 ha in area (Moreno-Casasola & Monroy, 2006). The climate is warm sub-humid, with an average temperature of 25 °C and an annual precipitation ranging from 1,300 to 1,500 mm. The rainy season occurs from June to September, the season of northerly cold fronts from October to January and the dry season from February to May (Moreno-Casasola et al., 1982; Rico-Gray, 1993; Díaz-Castelazo et al., 2004). The study was carried out in the six most representative vegetation types within La Mancha: (1) coastal dune scrub (CDS), (2) pioneer dune vegetation (PDV), (3) tropical deciduous forest (TDF), (4) tropical deciduous flooding forest and wetland (TDF-W), (5) tropical sub-deciduous forest in young soil (TSF-Y) and (6) tropical sub-deciduous forest in old soil (TSF-O) (for more details about the study site, see Martínez-Adriano, Aguirre-Jaimes & Díaz-Castelazo, 2016).

Sampling design

In each vegetation type, we established one transect with 20 to 30 permanent points located approximately 20 m apart. The studied area has a considerable small size for a natural reserve and it is a highly heterogeneous ecosystem with some vegetation types having very narrow distributions (at those sites multiple transects would not be independent). That is why it was not possible to have true replicates and we have only one transect per vegetation type. However, transect length at each site was considerable and our censuses have mainly an observational, not a test/experimental approach. More importantly, the methodology of permanent points that we used allowed us to have enough points to provide a good sampling effort within that habitat heterogeneity, which allowed us to register all butterfly-flowering plant interactions.

Taking these permanent points as the centre, we established round plots of radius five m (Franco-Gaona, Llorente-Bousquets & Shapiro, 1988). Within these plots (to avoid overestimation of richness and interactions, only 20 plots were randomly selected in those transects with more than 20 plots), we recorded the frequencies of all adult butterflies that foraged in the flowers over a period of 15 minutes at each observation point. No records of butterflies not foraging on flowers were considered given the fact that we were interested in their interactions with plants mediated by flowers, which was the goal of the network analysis. Butterfly sampling was conducted monthly (from March 2013 to February 2014) at 07h00 (before occurring the anthesis) in all of the vegetation types in order to record all possible species. In the vegetation types where tree cover is high, such as TSF-Y and TSF-O, TDF and TDF-W, only butterflies observed below the canopy were considered (plants and trees up to four meters in height).

Butterfly collection and identification

We collected butterflies directly from flowers using entomological nets. For each butterfly individual, we recorded the sampling point, plant species and vegetation type where individuals were collected, as well as morphological characters of each butterfly species (e.g., colour patterns and size, which helped us in the determination of each species). To preserve the butterfly specimens, we performed a standard Lepidoptera mount following the methods of Riley (1892) and Bland & Jaques (1978). Identification was carried out using field guides (Glassberg, 2007; Hernández-Baz et al., 2010; Díaz-Batres & Llorente-Bousquets, 2011), taxonomic keys (Bland & Jaques, 1978), official internet pages of butterflies and moths (https://www.butterfliesandmoths.org/; http://janzen.sas.upenn.edu/; http://www.butterfliesofamerica.com/) and through comparison with identified species deposited in the collection of the Museo de Zoología of the Facultad de Ciencias, UNAM.

Butterfly richness and diversity

Butterfly richness was expressed as the number of species present in each vegetation type. The butterfly richness included all individuals collected during all sampled months (we represented whole seasons in the samples). It is important to note that transects corresponding to TSF-Y and TSF-O were considered as only one tropical sub-deciduous forest (TSF) (described in Martínez-Adriano, Aguirre-Jaimes & Díaz-Castelazo, 2016) since the TSF-Y and TSF-O presented, in general, the lowest number of plant and butterfly species and had similar plant species composition. Furthermore, the vegetation types with the lowest diversity of plant-butterfly interactions were merged to avoid the effect of size on network metrics (Luna et al., 2017). Thus, these matrices could give inaccurate results in the calculation of network metrics. We calculated the Shannon Diversity Index (Hdiversity′) with the identity and presence of butterfly species per vegetation type. In order to determine which vegetation types were most similar in terms of butterfly composition, we grouped the habitats with a cluster analysis by means of Jaccard Similarity with Unweighted Pair Group Method with Arithmetic Mean algorithm (UPGMA), using Past software version 3.06 (Hammer, Harper & Ryan, 2001). We used a bootstrapping of 9,999 replicates to represent the percentage of replicates where each node is still supported given its clustering in the dendrogram (Hammer, Harper & Ryan, 2001). In addition, we calculated a qualitative Classic Jaccard Index for similarity (Jclas) in shared species with the presence of butterfly species per vegetation type (Chao et al., 2005) using EstimateS software V 9.1 (Colwell, 2013).

Network analysis

With the identity of species of plants and butterflies in the five transects, we constructed a matrix of interactions per vegetation type. In order to obtain the network-level estimates for the diversity of interactions, the frequency of plant-butterfly interactions in each vegetation type was used to calculate the Shannon diversity index of interactions (Hinteractions′) per vegetation type using the NETWORKLEVEL function of the BIPARTITE package (Dormann, Gruber & Fruend, 2008; Dormann et al., 2009; Dormann, 2011) on R console software (R Core Team, 2014).

In order to determine non-random patterns in the community-level network of plant-butterfly interactions in all of the vegetation types, we explored the existence of nested patterns at network-level structure. Nestedness consists of a few generalist species with a high number of links with other species, as well as specialist species that have few links or interactions that also regularly interact with the generalist species within the network of interactions. Finally, nestedness implies that there are very few or even no species-specific interactions and thus the species with few interactions (specialists) have links with the generalist core of the network (Bascompte et al., 2003; Bascompte, Jordano & Olesen, 2006). In order to determine whether there was an asymmetric structure in our network between butterflies and plants, we calculated the nestedness with NODF estimator of the ANINHADO software (Guimarães Jr & aes, 2006; Almeida-Neto et al., 2008). A nested network was considered where the NODF value observed for our network (NODF_total) was higher than predicted by the null model Ce (NODF(Ce)), with 1,000 randomizations for each network (Guimarães et al., 2006; Rico-Gray et al., 2012; Díaz-Castelazo et al., 2013). The null model Ce correspond to null model II of Bascompte et al. (2003). It assumes that the probability of an occurring interaction is proportional to the observed number of interactions of both trophic levels (Bascompte et al., 2003), in this case, plants and butterflies.

To fully explore the general network structure, with the BIPARTITE package (Dormann, Gruber & Fruend, 2008) on the R console software (R Core Team, 2014), we calculated the following network-level metrics (Dormann et al., 2009): (1) web asymmetry (WBAS), meaning the balance between numbers in the two trophic levels, where positive values indicate more higher-trophic level species, while negative values denote more lower-trophic level species, calculated as (ncol(web)-nrow(web))/sum (sum(dim(web)) (Blüthgen et al., 2007), (2) interaction strength asymmetry (ISA); this network parameter explained the dependence asymmetry and is also a measure of specialization, across both trophic levels, where positive values indicate higher dependence in the higher trophic level and negative values higher dependence in the lower trophic level (Blüthgen et al., 2007), and (3) specialization asymmetry (SA) for each guild (i.e., plants or butterflies), which is the average guild asymmetry of specialization, based on d′ (the specialization of each species based on its discrimination from a random selection of partners; for details, see Blüthgen, Menzel & Blüthgen, 2006). Since the mean d-value for the lower trophic level is subtracted from that of the higher trophic level, positive values indicate greater specialization of the higher trophic level (Dormann et al., 2009).

For network analyses at species level in the general network (Dormann, 2011), we calculated the following parameters: (1) species degree (DEG), which is the sum of links per species, (2) species strength (ST), which quantifies the importance of a species (i.e., as a resource or as a service provider) across all of its partners and is defined as the sum of dependencies of each species (Bascompte, Jordano & Olesen, 2006), and (3) interaction push/pull (IPP), which is the direction of interaction asymmetry based on dependencies; positive values indicate that a species more strongly affects the species of the other level with which it interacts than vice versa (“pusher”), negative values indicate that a species is, on average, more strongly affected by its counterpart (“being pulled”) (Vázquez et al., 2007). Jordano (1987) define the dependence as the relative interaction strength between two taxa. That is, the percentage of all interactions occurring between a particular pair of species. In addition, we defined core–periphery species of each species on the network, with a function on R software developed by CAMA based on the method proposed by Dáttilo, Guimarães & Izzo (2013), where the species with values >1 were core species and species <1 were peripheral species. Core species are those that interact with virtually all species in the network (Bascompte et al., 2003), while peripheral species are species with a lower number of interactions in relation to other species of the same trophic level (Dáttilo, Guimarães & Izzo, 2013)

In order to determine whether network parameters varied among vegetation types, we split the general interaction network into vegetation types. For each butterfly-plant interaction network, we calculated the following network traits: network size (the total number of species of both trophic levels recorded in each network), NODF, WBAS, ISA, SA, H′interactions, and core–periphery species of each species on the network.

The selection of all metrics was based on the information that each parameter provide for explain network structure and the importance of each species. We used the information provided by the technical manual of BIPARTITE package (Dormann, Gruber & Fruend, 2008) and the following references: Ings et al. (2009), Vázquez et al. (2009), Kaiser-Bunbury & Blüthgen (2015), Jordano & Stouffer (2016).

Results

From March 2013 to February 2014, we found 123 Lepidoptera species belonging to 87 genera (not including those butterflies unidentified to genus level) and 11 families (Table S1). Of these, 114 species were butterflies and nine were moths. Hesperiidae was the family with the highest number of species (46 species), followed by Nymphalidae (28 species), Pieridae (14 species), and Lycaenidae (13 species), while the other seven families had from one to seven species (Fig. S1A). The same order was observed in the highest number of genera per family (Fig. S1B). In terms of species number, four families (Hesperiidae, Nymphalidae, Pieridae, and Lycaenidae) comprised more than 82% of all the butterfly species observed and more than 79% of all the genera (complete list of butterfly species in Table S1).

The vegetation type with the highest number of species was pioneer dune vegetation (PDV; 75 species). The second most diverse vegetation type was the coastal dune scrub (CDS; 52 species), followed by tropical deciduous flooding forest and wetland (TDF-W; 51 species), tropical deciduous forest (TDF; 45 species) and tropical sub-deciduous forest (TSF; 8 species). For this latter vegetation type, it should be recalled that we merged the data from the TSF-Y and TSF-O since both transects of tropical sub-deciduous forest presented a low number of Lepidoptera species, as well as similarities in vegetation. According to the Shannon Diversity index, the vegetation type with the highest diversity of butterflies was PDV (H′ = 4.317), followed by CDS (H′ = 3.951), TDF-W (H′ = 3.932), TDF (H′ = 3.807) and TSF (H′ = 2.079). The cluster analysis based on Jaccard similarity index showed three groups of vegetation types, one with CDS, PDV, and TDF and the other two composed of just one vegetation type each (TDF-W and TSF, respectively) (Fig. 1). The vegetation types with most similarity in terms of species composition were CDS and PDV (Jclas = 0.427), CDS and TDF (Jclas = 0.386), PDV and TDF (Jclas = 0.29), and PDV and TDF-W (Jclas = 0.286). The vegetation type with the highest diversity of interactions was PDV (Hinteractions′=4.347), followed by TDF-W (Hinteractions′=3.972), CDS (Hinteractions′=3.747), TDF (Hinteractions′=3.61) and TSF (Hinteractions′=1.81).

Figure 1 Cluster analysis of vegetation types based on presence/absence of butterfly species in each vegetation type, using the Classic Jaccard Similarity Index.

CDS, coastal dune scrub, PDV, pioneer dune vegetation; TDF, tropical deciduous forest; TDF-W, tropical deciduous flooding forest with wetland; TSF, tropical sub-deciduous forest. The letters at the right side of the cluster mean the groups formed by similarity index (the cluster was calculated with the UPGMA algorithm). The number close to each branch means the percentage of replicates that each node is still supported is given on the dendrogram (according to Hammer, Harper & Ryan, 2001).

Butterfly-plant interaction network

We obtained 742 records of butterfly-flowering plant interactions. The interaction network consisted of 185 species (123 butterfly species-morphospecies and 62 plant species) with 3,010 interactions (Fig. 2A). The butterfly-plant interaction network was significantly nested (NODF_total = 15.98, NODF(Ce) = 8.94, P < 0.01). We observed that higher trophic levels had more species (WBAS = 0.329) and that the butterflies showed higher dependence than plants (ISA = 0.126), while the specialization asymmetry showed higher specialization of plants than butterflies (SA = −0.108).

Figure 2 Quantitative butterfly-flowering plant interaction networks.

Nodes on the left of each network are species of flowering plants, nodes on the right are species of butterflies. The green nodes represent peripheral plants, yellow nodes are peripheral butterflies and black nodes are core species within the guilds in each network (see definitions of core periphery species in the main text). The thickness of each link (gray lines) indicates the frequency of each pairwise interaction. (A) General interaction network; (B) network for coastal dune scrub; (C) network for pioneer dune vegetation; (D) network for tropical deciduous forest; (E) network for tropical deciduous flooding forest with wetland; and (F) network for tropical sub-deciduous forest. Abbreviations of all species in the networks are available in Data S1.

The analysis at species level of the butterflies (higher trophic level) showed Ascia monuste monuste and Euptoieta hegesia meridiania to be the species with the highest degree (both interacting with 15 plant species), followed by Leptotes cassius cassidula (interacting with 12 plant species). For plants (lower trophic level), the highest degree value was for Bidens pilosa (interacting with 50 butterfly species), followed by Lantana camara (28 butterfly species) and Ageratum corymbosum (25 butterfly species).

Horama oedippus was the butterfly species with highest species strength or importance (ST = 4.851), followed by Euptoieta hegesia meridiania (ST = 3.374) and Leptotes cassius cassidula (ST = 3.163). For plants, Bidens pilosa had the highest species strength (ST = 16.76), followed by Lantana camara (ST = 8.664) and Cordia spinescens (ST = 8.081). According to the interaction push/pull index, we observed that the species Horama oedippus (IPP = 0.385), Astraptes fulgerator azul (IPP = 0.221) and Dryas iulia moderata (IPP = 0.209) affected the plants with which they interact, while the other butterfly species were themselves affected by their interactions with plants. Plant species such as Cordia spinescens (IPP = 0.544), Lysiloma divaricatum (IPP = 0.395) and Thalia geniculata (IPP = 0.382) were those that most affected the butterfly visits. There are more species of plants that affect the butterfly visits; however, we noted that some core species (mainly in plant level) occurred only in a particular vegetation type.

Butterfly interaction network per vegetation type

We observed that the structure, descriptors, and parameters of the butterfly-flowering plant interaction networks (Fig. 2) varied among different vegetation types. We detected that some species observed as core to the general network of La Mancha also formed core components (core or central species) in the network analysis per vegetation type; this was the case with butterfly species such as Ascia monuste monuste, Euptoieta hegesia meridiania, Horama oedippus and Dryas iulia moderata, and the plant species Bidens pilosa, Lantana camara, Ageratum corymbosum, Turnera diffusa, Lysiloma divaricatum, Randia aculeata var. dasyclada and Sagittaria lancifolia (male flowers), which co-occurred as core components among networks in different vegetation types (Fig. 2). Nevertheless, we detected that some core species (mainly plants) per vegetation type were exclusive to a particular vegetation type.

We observed that butterfly interaction network descriptors and parameters varied among different vegetation types. With respect to network size, we observed that PDV had the highest number of species in the network (96 species), followed by CDS and TDF-W (both 71 species), TDF (59 species) and TSF (12 species) (only the latter two of these vegetation types were below the mean network size, Fig. 3A). In terms of the nestedness of each network per vegetation type, we observed significant nestedness for CDS (NODF_total = 24.7, NODF(Ce) = 15.29, P < 0.01) and PDV (NODF_total = 31.47, NODF(Ce) = 16.54, P < 0.01), while the non-nested networks were TDF (NODF_total = 17.57, NODF(Ce) = 16.54, P = 0.21), TDF-W (NODF_total = 14.40, NODF(Ce) = 11.47, P = 0.06) and TSF (NODF_total = 0.0, NODF(Ce) = 24.86, P = 0.96). The highest level of nestedness was found in PDV, followed by CDS (Fig. 3B). Regarding web asymmetry, we observed that all networks presented more butterfly than plant species since all of the WBAS values were positive. This pattern was also presented by the interaction strength asymmetry of each network and negative values for specialization asymmetry (Figs. 3C–3E). In the case of the Shannon diversity of interactions, PDV was found to be the most diverse vegetation type in interactions, followed by TDF-W, CDS, TDF and TSF, with the first four of these vegetation types more diverse than the mean value of interactions and only TSF presenting a value below that of the mean value of interactions (Fig. 3A).

Figure 3 Network parameters of the butterfly-plant interaction network.

The “X” axis of each graph shows the general interaction network followed by those of each vegetation type. CDS, coastal dune scrub; PDV, pioneer dune vegetation; TDF, tropical deciduous forest; TDF-W, tropical deciduous flood forest with wetland; TSF, tropical sub-deciduous forest. The parameters graphed for each interaction network were as follows: (A) network size, the points represent the number of species of each butterfly-flowering plant network; (B) total nestedness, the asterisk above the points represents the interaction networks that were significantly nested; (C) web asymmetry; (D) interaction strength asymmetry; (E) strength asymmetry and (F) Shannon diversity of interactions (H′). The gray line represents the mean value of each variable.

Discussion

In this study, we recorded 114 species of diurnal butterflies, that represent from 3 to 5.4% of all species recorded in Mexico (Llorente, Luis-Martínez & Vargas, 2006a; Hernández-Baz et al., 2010; Llorente-Bousquets et al., 2014), and 15.6% of the butterfly species in Veracruz (Llorente, Luis-Martínez & Vargas, 2006a; Hernández-Baz et al., 2010). The nine moth species we recorded foraging on plants represent 0.4% of the total number of species recorded for Veracruz (Hernández-Baz et al., 2010) and 0.1% of the total number of butterflies estimated for Veracruz (Becker, 2000; Hernández-Baz & Iglesias-Andreu, 2001). Our results pertaining to species number per family are similar to those reported for the biogeographic regions Nearctic, Neotropical, Palearctic, Ethiopic, Oriental (Hernández-Baz et al., 2010) and Australia-Oceania (Shields, 1989; Heppner, 1991), given that the four families with the highest number of species were Hesperiidae, Nymphalidae, Pieridae, and Lycaenidae. Furthermore, this is the same order observed for all Mexican butterfly species (Llorente, Luis-Martínez & Vargas, 2006a; Llorente et al., 2006b) and for Veracruz State (Llorente, Luis-Martínez & Vargas, 2006a). Our results are therefore in accordance with the general patterns cited above since Hesperiidae, Nymphalidae, Pieridae, and Lycaenidae accounted for more than 82% of species recorded in our study.

Diversity and similarity indices

Based on diversity indices of presence of species per vegetation type, we found that PDV, TDF-W, and CDS were the vegetation types with highest Shannon diversity index. This is similar to that found by Martínez-Adriano, Aguirre-Jaimes & Díaz-Castelazo (2016) in terms of the plant diversity index of the same vegetation types, where TDF was the vegetation type with highest diversity index, followed by PDV and CDS. This could be due to the fact that these vegetation types (apart from TDF-W) are open (present no canopy) and the butterflies can fly more easily than in closed or shaded vegetation types (e.g., TSF). Moreover, the floral resources that provide the main source of food for the butterflies (nectar and other liquids; Borror & White, 1970) are more available in the open areas of these vegetation types than in shaded areas.

Our results in terms of the similarity index show the highest similarities in butterfly species among CDS, PDV and TDF (first group), and two groups featuring one vegetation type each (TDF-W and TSF, respectively). This clustering pattern was also observed in other studies that involved plant species similarity in La Mancha, where similarities of CDS, TDF and TSF, could be due to sharing a similar sandy soil substrate as well as the close physical proximity of these vegetation types (Castillo-Campos & Travieso-Bello, 2006). In another study in La Mancha, the highest similarity of plant species floristic diversity among vegetation types was detected for CDS, TDF and PDV (Martínez-Adriano, Aguirre-Jaimes & Díaz-Castelazo, 2016). This pattern was also observed for CDS and PDV in the floristic diversity of plants with extrafloral nectaries at La Mancha, where the highest values of similarity were found in these two vegetation types (Díaz-Castelazo et al., 2004). These results are also similar to that found within the antagonistic system (plant-herbivorous caterpillar) studied by López-Carretero et al. (2014), since open areas were grouped into one cluster, in a manner similar to that found in a study of mutualistic interactions between ants and extrafloral nectaries (Díaz-Castelazo & Rico-Gray, 2015). Thus, the grouping pattern of butterfly species in the CDS, TDF, and PDV (driest vegetation types) could be due to similarities in the plant species present. The butterfly species could be responding to changes in food resources (e.g., number of flowers and nectar availability), showing that the butterfly species use the flower resources similarly in these three vegetation types. Many butterfly species recorded at TDF-W and TSF only occured within these vegetation types. This could be due to particular foraging habits or the fact that there are plants exclusive to these two vegetation types.

Interaction networks

In terms of the diversity of interactions of butterflies and plant species, we found high similarity with the calculated indices for butterfly species. This could be because the butterfly species were recorded only when they landed on flowers, and thus the species that presented no interaction with the plants (e.g., butterflies that simply flew through the plot or that landed in the plot, but not on the flower itself) are disregarded. However, only a few species of butterflies observed within the plots presented no interaction with the flowers (not listed in the present study), so possibly the indices of diversity of interactions do not vary significantly by including species not recorded using our methods. These results could indicate that most species of butterflies that appeared in La Mancha during this study do interact with the plants and therefore, in general, the butterfly species seem to present a high dependence on the flowers of the plants on which they feed.

On the other hand, our network of interactions was highly nested and only presented one module, which means that there were a few species with a high number of links with other species (generalists), as well as the specialist species that have few links or interactions that also regularly interact with the generalist species within the network of interactions (Bascompte et al., 2003; Bascompte, Jordano & Olesen, 2006). These results are in accordance with that stated in the literature for mutualistic systems (Díaz-Castelazo et al., 2010), which tend to have a nested structure and, to a lesser degree, a light modular structure. We also detected this pattern in CDS and PDV, since these networks presented the significant structure of nestedness, just as the general network of our study. This indicates that those interactions tend to be more generalist than specialist (as generally occur in butterfly species), and for this reason there may not be modules in our interaction network, since nestedness occurs when specialist species tend to interact with more generalist species (Bascompte et al., 2003; Almeida-Neto et al., 2008). Nevertheless, we did observe some modules in TDF, TDF-W, and TSF, but these were species-specific interactions forming isolated modules. A particular observation of such module formation took place in TSF since the lowest number of records of butterfly-plant interactions was found in this vegetation type. These low records in TSF could be due to the fact that most flowering species of this sub-deciduous forest occur above the canopy (where we did not record the interactions). In addition, the occurrence of nestedness in the general network and less modularity (without modules), suggests a lack of opportunism and/or specialized interactions among the species (or specialization of certain butterfly species). Therefore, these butterflies as flower visitors that may well benefit the plants visited since (at a community level), this interaction system potentially reflects the typical patterns of mutualistic networks, as is the case in pollination networks (Bascompte et al., 2003; Guimarães et al., 2006) where the general structure is more nested than modular.

The results of the core–periphery analysis showed that the core species in the general network of La Mancha, as with the core components of the network analyses per vegetation type, include the butterfly species Ascia monuste monuste, Euptoieta hegesia meridiania, Horama oedippus, and Dryas iulia moderata, and the plant species Bidens pilosa, Lantana camara, Ageratum corymbosum, Turnera diffusa, Lysiloma divaricatum, Randia aculeata var. dasyclada, and Sagittaria lancifolia (male flowers). Likewise, some core species such as the butterflies Horama oedippus, Euptoieta hegesia meridiania and Leptotes cassius cassidula and plants B. pilosa, L. camara, and Cordia spinescens present high values of Species Strength. The interactions of the push pull index showed the same trend since some butterfly species such as H. oedippus, Astraptes fulgerator azul, and D. iulia moderata affected the plants with which they interact; while the plants C. spinescens, L. divaricatum, and Thalia geniculata were found to affect the butterfly visits. This is of particular interest, because the species that we observed as core species has great importance to the structure of the networks of each vegetation type, while the species with highest values of species strength or with positive IPP values could be more important to the entire interaction network. These results showed that some species were important to the interaction networks, in terms of both their structure and ecological implications; this was the case for H. oedippus, one of the most important species according to the three aforementioned network parameters, which demonstrates the importance of that species to the whole community.

It appears that the butterfly species recorded in the present study did not show a particular preference for a specific group of flowers; i.e., we did not observe a modular pattern in our quantitative interaction networks. This pattern could be due to the opportunistic trait present in the butterflies group since they tend to visit a great diversity of flowers (colors, shapes, and sizes) in order to collect nectar. On most of their visits, this group of flower visitors could not contribute to the pollination process, except when visiting flowers that were adapted to the butterfly pollination syndrome. These results are similar to those found by Hernández-Yáñez et al. (2013) since they conclude that butterflies (and other floral visitors) show no particular preference for a specific flower color or shape in as much as this floral visitor guild visited most flowers/colors/shapes considered in that study.

With respect to the ISA and SA network parameters, our results showed that butterflies depend more on plants than vice versa. This may be because floral nectar forms an essential part of the diet of these insects (Borror & White, 1970), while plants can have floral visitors other than the butterflies (this pattern was observed for both the overall network and the sub-networks per vegetation type). As for the SA (and for all networks in general), it should be noted that the fact that the plants are more “specialized” than butterflies does not imply that the network presents true specialization since no modules were detected. The finding that the network has an asymmetry in specialization (more towards the plants) may be due to the fact that plants are sessile and spatially related to certain habitats, while butterflies may have a greater range in their interaction mobility. Another possible explanation for this specialization in plant group is that the numbers of potential floral visitors are underestimated because there could be others than the butterfly group. Thereby, the plant species (as feeding resource), could be limit the presence of butterflies.

Conclusions

This study provides the first characterization of the butterfly diversity and its interactions with flowering plants per vegetation type from the data of monthly censuses taken over the course of one year in the study area. Our study is important because it constitutes a pioneer study of butterfly-plant interactions in this natural protected area, at least for adult butterflies, that shows the diversity of interactions between flowering plants and butterflies as flower visitors. In addition, our research constitutes the first approach (at a community level) to explore the functional role of the pollination services that butterflies provide to the plant communities. We highlighted that open areas show a higher diversity and these areas shared a higher number of species that shaded sites. In the interaction networks parameters, our results highlighted the higher dependence of butterflies by the flowers on which they feed than vice versa. In conclusion, the plant species (as a feeding resource) seem to limit the presence of butterfly species. Thus, this protected area is highly relevant for Lepidoptera diversity and the interaction between these insects and flowering plants. We suggest that studying plant and butterfly diversity in tropical habitats will provide insight into their interspecific interactions and community structure.

Supplemental Information

Table S1 List of butterfly species per vegetation type and common plant species on which they feed

CDS, coastal dune scrub; PDV, pioneer dune vegetation; TDF, tropical deciduous forest; TDF-W, tropical deciduous flooding forest and wetland; TSF, tropical sub-deciduous forest. The nomenclature is based on official web pages (https://www.butterfliesandmoths.org/; http://janzen.sas.upenn.edu/; http://www.butterfliesofamerica.com/).

Click here for additional data file.

Figure S1 Taxonomic representativeness of butterfly species in La Mancha

(A) Number of species per family, Nymphalidae, Hesperiidae, Pieridae, and Lycaenidae comprised 82.11% of all species, while the remaining 17.89% comprised four families with seven (or less) species per family. (B) Number of genera per family. We observed the same trend with Nymphalidae, Hesperiidae, Pieridae and Lycaenidae for number of genera, since these four families comprised 79.31%. The numbers of species and genera is shown in parentheses.

Click here for additional data file.

Data S1 Quantitative matrices of plant-butterfly interactions networks

We provide all matrices and abbreviations of all species in the networks.

Click here for additional data file.

We give special thanks to Postgraduate Secretary of the Instituto de Ecología, A. C. and to the Department of Multitrophic Interactions (Red de Interacciones Multitróficas) for the administrative facilities provided to perform the present study. In addition, we give special thanks to Elisa Zaragoza, J. Luis Sánchez and Rosalba Quintana for their assistance with fieldwork. We also give special thanks to Fernando Aguilar, Anastacio García and Enrique López (La Mancha staff), for help and company in the biological station during all of the fieldwork visits. Keith MacMillan revised the English text of the manuscript. We give special thanks to two anonymous reviewers and the academic editor for help to improve this manuscript.

Additional Information and Declarations

Competing Interests

Author Contributions

Data Availability

The authors declare there are no competing interests.

Cristian A Martínez-Adriano conceived and designed the experiments, performed the experiments, analyzed the data, contributed reagents/materials/analysis tools, prepared figures and/or tables, authored or reviewed drafts of the paper, approved the final draft, wrote the paper.

Cecilia Díaz-Castelazo conceived and designed the experiments, contributed reagents/materials/analysis tools, authored or reviewed drafts of the paper, approved the final draft, wrote the paper.

Armando Aguirre-Jaimes conceived and designed the experiments, approved the final draft, wrote the paper.

The following information was supplied regarding data availability:

The raw data are provided in the Supplemental Files.

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
