# Peer review of "Flower-mediated plant-butterfly interactions in an heterogeneous tropical coastal ecosystem"

_PeerJ, doi:10.7717/peerj.5493_

## Round 0.1 · original submission · Major Revisions

· Academic Editor

Major Revisions

Yours ms has now been reviewed by two referees who acknowledge the interest of your work but find that the ms needs a major revision before it can be accepted for publication. I have looked at the ms myself, although not as deeply as them, and I agree with most of their concerns.
Basically, I think you need to re-write the entire ms, and shorten much of both the introduction and the discussion sections. Focus on your specific question in the introduction. Regarding your methods’ section, you need to clearly state how you sampled the interactions, and justify the metrics you used to describe network structure as well as the null models to test their significance. Please also state here whether you can consider your vegetation types as a gradient, as it is true that vegetation types and sites can be confounded if there are no replicates of the former. You also need to shorten and re-structure the discussion section, focusing on the results and making them more general. This is not a review article, however, and thus such a long list of references is not justified. Please refer only to those that are essential for your specific study. You also need to re-write figure legends including all information requested by ref 1.

Reviewer 1 ·

Basic reporting

The paper is well written using technically correct English. The introduction is sufficiently long to show the field of knowledge, and the relevant literature is cited. However, as detailed in the criteria on Experimental design, the introduction requires a thorough review.
The structure of the article conforms to PeerJ standard sections.
Figures:
Fig. 1: Both number of species and genera show similar patterns. The taxonomic representativeness is not a relevant point of the manuscript and it could be summarised in just one figure in the supplementary material (in this case, number of species).
Fig. 2: Which were the criteria for the species selection? If pictures of butterflies are to be included, they should be on relevant species based on criteria mentioned in the manuscript (e.g., degree, strength... or something like that)
Fig. 3: The dendrogram from cluster analysis on presence-absence of butterfly species represents three groups, but this is not clear at all. Which is the criterion for separating TDF-W from the group formed by TDF, PDV and CDS?
What do the figures next to dendrogram branches mean?
Fig. 4: Please, define core and peripheral species here or at least provide some reference to the main text. Provide also a reference to species abbreviations and use a larger font.
Fig. 5: Y-axis titles are missing from some panels (A, C and E).
The raw data is available from an Excel file. Maybe an extra sheet including the meaning of species abbreviations would be useful.

Experimental design

This manuscript presents original primary research on butterfly communities in a tropical region and their interactions with flowering plants used as nectar sources. My major concern on this work is that the research question is poorly defined. The introduction is too long and focuses on the scarcity of butterfly studies over three paragraphs (lines 52-90). I would recommend the authors to shorten the introduction and to be much more specific in the identification of the knowledge gap covered by their work. From my point of view, a good possibility would be focus on butterfly-nectar plant interactions, including community research as a secondary point, and just briefly mentioning the faunal point.
Other important concern relates to experimental design. One of the points addressed in the study is the effect of vegetation type (5 categories) on butterfly communities and network structure. Apparently, vegetation types are not replicated and any conclusion derived on this subject has the problem of being based on confounding effects (vegetation types and sites). This limitation could be overcome if the vegetation types represented a continuum along a gradient, but I do not know if this is the case.
Minor points
Lines 161-162: the sampling was restricted to butterflies foraging on flowers. This is a limitation (discussed in the manuscript) for the research on community characterization, but not on network structure.
Please, provide here the year and the period of sampling.
Lines 182-183: what is the actual sample? Are all individuals collected within a site over the whole season?
Lines 192-198: “With the identity…” These sentences should be moved to the Network analysis section.
Lines 199-212: I understand that nestedness is also network level estimate.
Lines 209-211: please, provide further details on null model 2.
Lines 199-243: which are the bases for selecting the indices at network and species level used in the manuscript?
Lines 230-232: please, provide further details on “dependencies”.
Lines 240-243: the “core-periphery species” issue is cited here by first time.

Validity of the findings

The discussion is too long. The contents in lines 338-385 could be much shortened by summarizing all faunal information.
The conclusions should not start with the list of butterflies in La Mancha, because it sounds too local.
Minor points
Line 246: “Lepidoptera species” instead of “butterfly species”.
Lines 251-253: the sentence concerning the number of genera could be removed according to the points raised for Fig. 1.
Lines 275-279: the passage referring to the similarity and cluster analyses should be revised according to the points suggested for Fig. 3.
Line 282: some individuals were not identified to species.
Lines 283-285: please, move the results concerning differences between vegetation types to the corresponding section.
Line 286: please, include further details on the meaning of NODF_total and NODF(Ce).
Lines 330-331: plants were more specialized than butterflies according to the SA index (negative values).
Lines 487-492: as stated in line 486, one possible explanation for the larger specialization of plants is that the numbers of potential floral visitors are underestimated because there could be others than butterflies.
Lines 492-494: this is an unclear sentence and requires some rewriting.

Additional comments

Abstract
Line 21: the study area is not mentioned in the methods subsection.
Lines 26-34: the network results are barely explained in the abstract.
Lines 43-45: the final conclusion sounds too general.

Reviewer 2 ·

Basic reporting

This MS explores the relationships between adult butterflies and their nectar resources in a set of plant communities in a subtropical coastal area in Mexico. Based on timed observations of nectaring butterflies in circular plots in each of 5 habitat types, the authors carried out an analysis of the mutualistic networks between butterflies and flowering plants. Different measures of network structure were used for a comparison of the different communities.

Network data on interactions between adult butterflies and flower species are scarce, even in temperate ecosystems. In this respect, the MS gives interesting and novel information, more so representing a tropical system. Moreover, sampling was made year-round, in one year, totalling over 700 butterly-plant interactions, which represents a rather important dataset.

In spite of this, I believe that the MS needs major rewriting before being considered for publication. In particular, I see some aspects that need major improvement:

1 - In general, the MS it's too long and must be shortenend. For instance, literature is too extensive (ca. 90 references!) and comprises many local titles that to me are too detailed (e.g. local lists of Mexican butterflies). I would keep only those essential and comprehensive works that may be of general interest for non-local entomologists.

2 - Many information given in the Results section could appear in tables, which would not only help to shorten the presentation but also to make it more understandable for the reader. For instance, data on diversity and network metrics (l. 266-355) could be much condensed on a couple of tables.

3 - I would skip Fig. 1 and avoid much of the descriptive results in l. 246-265, which are too detailed and of little interest for the general reader. The same for the first part of Discussion.

On the other hand, I missed some more deep arguments of the expected findings (l. 136-143), which could be somewhat extended and more based on previous findings by other researchers, and then more related to the Discussion.

Experimental design

I don't see the point of distinguishing habitat categories 5 and 6 (l. 154-155), and then pool the data from them into a single category (l. 183-185). Moreover, doing so, you're oversampling this final category, even if diversity here is much lower than on other habitats.

Validity of the findings

No comment.

Additional comments

See some more specific comments on various parts of the MS:

- The expression 'host plant' when referring to nectar sources is probably not correct. I would say that 'host plant' is used only for the tight relationship between larvae and the plants they feed on. In the current context, I would it replace by 'adult resources' or similar expressions.

- The writing should be improved in some parts of the MS. Just some examples, that at present consist of sentences or arguments that are difficult to follow: l. 64-67, l. 94.97, , l. 142-143, l. 364-365, l. 448-453.

- I don't see the point of giving 2 decimals when discussing percentage of species found in the study area compared with local lists. I even would skip these figures, which are of relatively little interest.

- Some references are mising in the references sections.

---

## Round 0.2 · Major Revisions

· Academic Editor

Major Revisions

Your revised ms has been sent to the most critical reviewer who acknowledges the changes you have made to improve the last version. However, this reviewer still points out a number of concerns, although I think they are minor, that you should address before I can deem your ms as publishable.

Please carefully check the style and writing before your re-submit the ms.

I look forward to receiving your revised ms in due course.

Sincerely,
Anna Traveset

Reviewer 1 ·

Basic reporting

I much appreciate the author's work for improving their manuscript. They have corrected the figures and moved those of less importance to supplementary material. They have also incorporated most comments to the main text. However, I am afraid there was a misunderstanding concerning the comments to fig. 3. It is clear that the metric used for the dendrogram was Jaccard similarity index. The point raised in the review was actually about the algorithm used for constructing the dendrogram and the threshold to establish groups, not about the metric. Another misunderstanding concerns the figures next to dendrogram branches: I referred to figures 100, 99, 89 and 72.

Experimental design

The introduction is still too long and focused on butterflies and the study area, it sound too local. For instance, the text in lines 55-111 (line numbers based on the document with track changes for all comments) could be substantially shortened.
With respect to the experimental design, the problem associated to the lack of replicated vegetation types is not clearly explained in the text included (lines 181-185).
Index selection (lines 234-298): it could be useful to include some key references.

Validity of the findings

The discussion is still too long. I feel that is still possible to condense the passage concerning the faunal information (lines 401-438). If possible, shortening of other discussion sections could much improve the ms.
The statement raised in lines 562-565 is somewhat circular: vegetation types are based on plant species, so they cannot limit their distribution. This is also included in the abstract (lines 48-52) and the conclusions (lines 576-582) and it should be revised.

Additional comments

Abstract: the background (lines 13-21) should be more focused on community ecology and particularly on pollinator networks, rather than on butterflies.
Conclusions (lines 567-582): I would recommend the authors to carefully think about the points to be included here. At the current stage, it sounds too vague and unfortunately does not provide much to the ms.
Please, check carefully the writing of the ms: at least, passages in lines 143-145, 163-164, 218-221 and 516-520 require some rewriting.

---

## Round 0.3 · accepted · Accept

· Academic Editor

Accept

I thank you for considering all points raised by reviewers which I think have been very useful to improve the quality of your work. I can deem now your paper as publishable in PeerJ.

Congratulations for this nice ms.

Kind regards,
Anna Traveset

#